# Multi-Objective Value Iteration with Parameterized Threshold-Based Safety Constraints

## Abstract

We consider an environment with multiple reward functions. One of them represents goal achievement and the others represent instantaneous safety conditions. We consider a scenario where the safety rewards should always be above some thresholds. The thresholds are parameters with values that differ between users. We efficiently compute a family of policies that cover all threshold-based constraints and maximize the goal achievement reward. We introduce a new parameterized threshold-based scalarization method of the reward vector that encodes our objective. We present novel data structures to store the value functions of the Bellman equation that allow their efficient computation using the value iteration algorithm. We present results for both discrete and continuous state spaces.

## 1 Introduction

In reinforcement learning (RL), we often face scenarios where one objective conflicts with safety constraints. For example, Pineau et al. (2007); Rush et al. (2004); Zhao et al. (2009) have used RL in controlled trial-based treatment analysis. In each treatment stage a medication is administered after which tests are performed to measure symptoms and side effects, and the objective of the doctor may be to optimize treatment for the symptoms while keeping the side effects below some threshold. Different patients may have different preferred thresholds, and hence, it is necessary to generate a family of treatment plans that covers all preferences. One can think of these preferences as values of parameters for threshold-based safety constraints.

We present a value iteration algorithm to compute optimal policies that satisfy safety constraints. We consider a scenario where the safety constraints are parameterized, and the computed family of policies has to cover all possible parameter values. Specifically, we consider an environment that provides the agent with a reward vector in $\mathbb{R}^{d+1}$ corresponding to $d+1$ reward functions after doing an action $a$ at a state $s$ and time $t$. We consider a discrete and finite action space $A$ and time horizon $T$. The state space $S$ can be finite or not. Both cases are discussed separately. The aim of the agent is to maximize the sum over time of the $(d+1)^{st}$ coordinate of the reward vector while making sure the other $d$ coordinates pass their thresholds at each time step. The thresholds are encoded in a vector $\delta \in \Delta := \mathbb{R}^d$. To do that, at each time step, we map the reward vector to a scalar equal to its $(d+1)^{st}$ coordinate when the thresholds are met and to negative infinity otherwise. Hence, the scalarized reward is parameterized by $\delta$.

We present two data structures DecRect and ContDecRect to store the $Q$ functions of the Bellman equation for discrete and continuous state spaces, respectively. They utilize the structure of the scalarized reward function for any state and action as a constant over an axis-parallel hyperrectangle in $\Delta$ with bottom left corner at $[-\infty]^d$ and $-\infty$ elsewhere. We are abusing notation here since $[-\infty]^d$ is not a point. This geometrical perspective to the problem inspires intuitive and efficient algorithms to find the pointwise maximum, sum and non-negative scalar multiplication of such functions. These three operations are the only ones needed in the computations of the Bellman equation. We show that they can be done by computing intersections of hyperrectangles in $\Delta$. Hence, in discrete state spaces, we represent any $Q$ function for a given state and action by a DecRect which is a set of pairs of hyperrectangles in $\Delta$ and values in $\mathbb{R}$. In continuous state spaces, we assume linear dependency of the reward functions on a feature space of the states of dimension $c$. Then, we represent the $Q$ function for a fixed action as sets of $d+1$ vectors that when multiplied by the feature vector of the state, provide a set of hyperrectangles and values from which the value of the

function can be inferred. In both discrete and continuous state space cases, the $Q$ function will be non-increasing in each dimension and hence the names DecRect and ContDecRect for *decreasing*, *rectangles* and *continuous*.

Using DecRect, for finite state spaces, we present an algorithm that uses polynomial time and space in $|S|$ and $|A|$, linear in $T$ and exponential in $d$, to compute a family of policies that cover all possible values of $\delta$. Moreover, we present an efficient method to identify dominated actions using Pareto front computations. For continuous state spaces, we provide an algorithm that uses ContDecRect and with time and space complexity that is exponential in $T$ and polynomial in $|A|$, $d$ and $c$, to compute a similar family of policies.

The paper is inspired by the work: "Efficient Reinforcement Learning with Multiple Reward Functions for Randomized Controlled Trial Analysis" by Lizotte et al. (2010) and its extended version Lizotte et al. (2012). That work aims to maximize the sum over time of a weighted average of the reward vector components instead of a single component with constraints on the others. They assume finite time horizon $T$ and unknown weights, too. They provided an efficient algorithm, yet exponential in $T$, to synthesize a set of optimal policies that cover all possible weights for cases with up to three reward functions. For more than three, their time complexity bound becomes doubly exponential. There is rich literature focused on multi-objective RL with constraints. For example, Gábor et al. (1998) present a lexicographical order on value functions of policies represented as pairs of reals while having a constraint on the first component. They provide an algorithm to learn an optimal policy based on that order. There are multiple differences with our formulation: their constraint on the first reward is on the total expected reward of the policy while ours is on multiple immediate (instantaneous) rewards. Second, they synthesize a single policy while we generate a family of policies that covers all possible thresholds $\Delta$. Finally, they consider an infinite time discounted reward scenario, while we consider a bounded time scenario. A nice overview of existing algorithms for multi-objective reinforcement learning can be found in Roijers et al. (2013).

Our contributions can be summarized as follows:

- A threshold-based scalarization function of the reward vector that encodes the goal achievement objective and the safety constraints. It inspires geometric perspective that allows efficient computation and representation of policies.

- Efficient data structures to store the $Q$ and $V$ functions of the Bellman equation for both continuous and discrete state spaces.

- Efficient algorithms that construct a family of policies for discrete and continuous state spaces with arbitrary number of features that cover a set of safety preferences encoded as threshold vectors.

We start by defining some notations in Section 2. Then, we formally describe our setup and problem statement in Section 3. Section 4 presents the data structure DecRect and a value iteration algorithm to compute a family of policies for all possible thresholds in the case of discrete state space. After that, Section 5 present similar results for the case of continuous state spaces using the data structure ContDecRect. Finally, we conclude and suggest future work in Section 6.

## 2 PRELIMINARIES

For any positive integer $d$, we denote the set $\{1, \ldots, d\}$ by $[d]$. The natural partial ordering on vectors in $\mathbb{R}^d$ is defined as follows: $\forall x, x' \in \mathbb{R}^d$, $x \leq x'$ if and only if $x[i] \leq x'[i]$ for all $i \leq d$. Given any two matrices $X_1 \in \mathbb{R}^{d_1 \times d_2}$ and $X_2 \in \mathbb{R}^{d'_1 \times d_2}$, we denote by $[X_1, X_2] \in \mathbb{R}^{(d_1 + d'_1) \times d_2}$ the matrix that results from concatenating $X_1$ and $X_2$. Similarly, given any two matrices $X_1 \in \mathbb{R}^{d_1 \times d_2 \times d_3}$ and $X_2 \in \mathbb{R}^{d'_1 \times d_2 \times d_3}$, we denote by $[X_1, X_2] \in \mathbb{R}^{(d_1 + d'_1) \times d_2 \times d_3}$ the matrix that results from concatenating $X_1$ and $X_2$. Given a finite set $S$, we denote the cardinality of $S$ by $|S|$.

We define a special class of functions and some of its properties.

**Definition 1.** *A function $f : \mathbb{R}^d \to \mathbb{R}$ said to be piecewise constant function (PWC) if there exists finite number of disjoint axis parallel hyperrectangles, possibly with corners at infinity, over which the value of $f$ is constant . If $f$ is PWC and non-increasing along each dimension, it is said to be non-increasing PWC (NIPWC).*

The classes of PWC and NIPWC functions are closed under addition, non-negative scalar multiplication, and pointwise maximization. That is, given PWC functions $f_1$ and $f_2 : \mathbb{R}^d \to \mathbb{R}$, the functions $f_{max}(\delta) := \max\{f_1(\delta), f_2(\delta)\}$ and $f_{sum}(\delta) := f_1(\delta) + f_2(\delta)$, for $\delta \in \mathbb{R}^d$ are also PWC. Moreover, for any $\alpha \in \mathbb{R}$, $\alpha f_1$ is PWC. If $f_1$ and $f_2$ are NIPWC and $\alpha \geq 0$, then $f_{max}, f_{sum}$, and $\alpha f_1$ are NIPWC.

## 3    SETUP

We consider an *agent* operating in an *environment* over a finite time horizon $T$. The state and action spaces are $S$ and $A$. If the agent takes action $a_t \in A$ at state $s_t \in S$ and time $t \leq T$, then the environment returns a $(d + 1)$-dimensional *reward* vector $r_t(s_t, a_t) \in \mathbb{R}^{d+1}$. The first $d$ components of the reward are used to specify *safety constraints* and the $(d + 1)^{st}$ component corresponds to achievement of a goal. For a parameter $\delta \in \Delta$, our aim is to maximize the sum of the last component of the reward over the time horizon while keeping the other $d$ components above their correspondent thresholds specified by $\delta$. Our objective is to design a parameterized policy $\pi_\delta$ that solves the problem for any constant $\delta \in \Delta$.

Formally, we consider a Markov Decision Process (MDP) $M_\delta$ parameterized by $\delta \in \Delta$ with state space $S$, finite action space $A$, state transition matrix $P$ and a reward function $r_t : S \times A \to \mathbb{R}^{d+1}$. Using $r_t$ we define the *scalarized reward* $R_t$ at time $t$ defined as follows: for any $s_t \in S$ and $a_t \in A$,

$$R_t(s_t, a_t, \delta) := \begin{cases} r_t(s_t, a_t)[d + 1], & \text{if } \forall\, i \leq d,\ \delta[i] \leq r_t(s_t, a_t)[i], \text{and}, \\ -\infty, & \text{otherwise.} \end{cases} \tag{1}$$

A policy $\pi$ for $M_\delta$ maps a time point and a state to an action, that is, $\pi : [T] \times S \to A$.

**Problem Statement**  The aim is to design a family of policies $\Pi := \{\pi : [T] \times S \to A\}$ such that for any $\delta \in \Delta$, we can find a corresponding optimal policy $\pi_\delta \in \Pi$ that maximizes $\mathbb{E}[\sum_{t=1}^{T} R_t(s_t, a_t, \delta)]$.

For any $t \leq T$, we let $Q_t : S \times A \times \Delta \to \mathbb{R}$ be the optimal $Q$-function and let $V_t : S \times \Delta \to \mathbb{R}$ be the *value* function at the $t^{th}$ time step. For a given value of the parameter $\delta \in \Delta$, $Q_t(s_t, a_t, \delta)$ is the total reward if action $a_t \in A$ is taken at state $s_t \in S$ and the optimal policy is followed till time $T$. Similarly, $V_t(s_t, \delta)$ is the total reward if the optimal policy was followed from the time step $t$ till time $T$ starting from state $s_t \in S$.

For a terminal state $s_T \in S$ and action $a_T \in A$,

$$Q_T(s_T, a_T, \delta) := R_T(s_T, a_T, \delta) \tag{2}$$

is the terminal reward. Similar to Lizotte et al. (2010), we will present a value iteration algorithm which computes the $Q_t$ and $V_t$ functions for all possible inputs recursively starting from $T$ and backwards using the Bellman equation: for any $\delta \in \Delta$, $t \leq T$, $s_t \in S$, and $a_t \in A$:

$$Q_t(s_t, a_t, \delta) = R_t(s_t, a_t, \delta) + \mathbb{E}_{s_{t+1}|s_t, a_t}[V_{t+1}(s_{t+1}, \delta)], \tag{3}$$

where $V_t(s_t, \delta) := \max_{a \in A}(Q_t(s_t, a, \delta))$.

As usual, once we evaluate the $Q$-functions, we can compute the optimal policy for a given $\delta \in \Delta$ by applying the action $\arg\max_{a \in A} Q_t(s_t, a, \delta)$ at state $s_t \in S$ and time $t \leq T$.

## 4    DATA STRUCTURE FOR PARAMETERIZED VALUE FUNCTIONS: DISCRETE STATE SPACE

In this section, we consider a discrete state space $S$ and show how to compute $Q_t$ and $V_t$. We start by introducing the data structure *DecRect* for representing these functions in Section 4.1. This representation is key for improving the efficiency of computing maximization and expectation in Equation (3), as we will discuss later in Sections 4.2, 4.3, and 4.4. We use the following example to explain the different concepts in this section.

**Example**  We borrow the example shown in Figure 1a from Lizotte et al. (2010). The MDP consists of a single state $s_T$ and four actions $a_1, a_2, a_3$ and $a_4$ with 2-dimensional reward vectors, and hence,

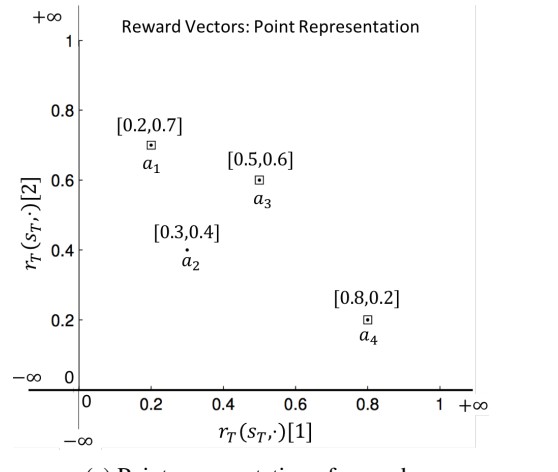
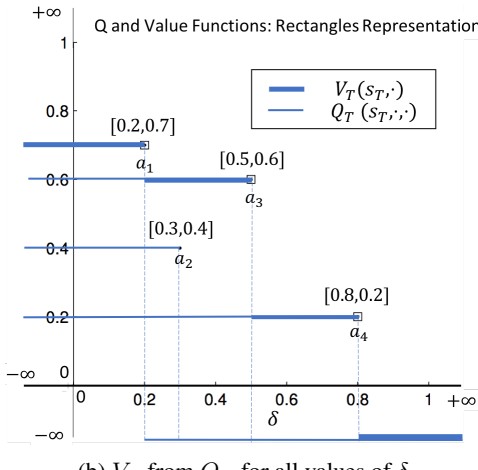

(a) Point representation of rewards     (b) $V_T$ from $Q_T$ for all values of $\delta$

Figure 1: Example with four actions with the corresponding reward vectors, $Q_T$ and $V_T$

$d = 1$. These vectors are represented as points in the plane. From this representation, we can get the representation of the scalarized terminal reward function for each of the actions. In Figure 1b, we plot the $Q_T$ function for each action and $V_T$ versus $\delta \in \Delta = \mathbb{R}$. As noted earlier, $Q_T = R_T$ which is piecewise constant and shown in the thin lines in the figure and $V_T$ is a pointwise maximization of such functions and shown in the thick lines. The points corresponding to actions that are optimal for some $\delta$ are surrounded by a square. That is why the one corresponding to action $a_2$ is not.

### 4.1 DecRect Representation of NIPWC functions

First, observe that for any fixed $s_t \in S$ and $a_t \in A$, $R_t(s_t, a_t, \cdot)$ is NIPWC of $\delta \in \Delta$ as it takes the value of $r_t(s_t, a_t)[d+1]$ or $-\infty$. It is a positive constant over axis parallel hyperrectangle $\delta[i] \leq r_t(s_t, a_t)[i]$ for $i \leq d$, and $-\infty$ elsewhere. By Equation (2), it follows that $Q_T(s_T, a_T, \cdot)$ has the same properties as $R_T(s_T, a_T, \cdot)$ (Figure 1b). For example, the $Q$-functions in Figure 1b are positive over intervals that start from $-\infty$, they are non-increasing and piecewise constant.

Moreover, observe from Equation (3) that the operations for computing $V_t$ and $Q_t$ consists of a combination of maximization, non-negative scalar multiplication, and addition of NIPWC functions, which means that these functions are always NIPWC.

We will represent an NIPWC function $f : \Delta \to \mathbb{R}$ by a set of $m > 0$ hyperrectangles in $\Delta$ and the corresponding values in $\mathbb{R}$; $f(\delta)$ is then defined as the maximum value over all the hyperrectangles that contain $\delta$. Assuming that each hyperrectangle has the bottom-left corner at $[-\infty]^d$, it can be represented only by its top-right corner in $\Delta$. Collecting all $m$ vertex-value pairs, we represent $f$ by a matrix: $X = [x_1, \ldots, x_m]^\intercal$, where each row $x_j$, $j \leq m$, is a pair in $\Delta \times \mathbb{R}$. We call this representation DecRect. The semantics of DecRect is as follows: for any $\delta \in \Delta$, let $J_\delta = \{j : \forall i \leq d, \delta[i] \leq X[j][i]\}$, then

$$f(\delta) := \begin{cases} \max_{J_\delta} X[j][d+1], & \text{if } |J_\delta| \neq 0 \\ -\infty, & \text{otherwise.} \end{cases} \tag{4}$$

To reiterate, each row $X[j]$ is $(d+1)$-dimensional; the first $d$-components represent the upper-right corner of a hyperrectangle, and $X[j][d+1]$ is the corresponding value.

**Example.** The DecRect representation of the NIPWC function $Q_T(s_T, a_1, \cdot)$ of Figure 1b will be: $[[0.2, 0.7]]$, $Q_T(s_T, a_2, \cdot)$ will be $[[0.3, 0.4]]$, $Q_T(s_T, a_3, \cdot)$ will be $[[0.5, 0.6]]$ and $Q_T(s_T, a_4, \cdot)$ will be $[[0.8, 0.2]]$. $V_T(s_T, \cdot)$ will be represented as $[[0.2, 0.7], [0.3, 0.4], [0.5, 0.6], [0.8, 0.2]]$ or $[[0.2, 0.7], [0.5, 0.6], [0.8, 0.2]]$, as $a_3$ is not optimal for any $\delta$. We will discuss the last point more in Section 4.3.

## 4.2 MAXIMIZATION OF NIPWC FUNCTIONS

In this section, we describe how to maximize two NIPWC functions represented by two DecRects. Assume we are given two functions $f_1$ and $f_2 : \Delta \to \mathbb{R}$ represented by two DecRects $X_1$ and $X_2$. The pointwise maximum $f_{max}$ of $f_1$ and $f_2$ is represented simply as $X_{max} = [X_1, X_2]$. This representation is indeed correct because:

$$f_{max}(\delta) := \max\{f_1(\delta), f_2(\delta)\} = \max_{j,h: \; h \in \{1,2\} \text{ and } \forall i \leq d, \; \delta[i] \leq X_h[j][i]} X_h[j][d+1] \qquad (5)$$

$$= \max_{j: \; \forall i \leq d, \; \delta[i] \leq X_{max}[j][i]} X_{max}[j][d+1], \qquad (6)$$

while equal to $-\infty$ when neither $X_1$ nor $X_2$ has hyperrectangles that contain $\delta$. For any given $\delta \in \Delta$, $f_{max}(\delta)$ is equal to the maximum of the values of the functions on all hyperrectangles that contain $\delta$ and equal to $-\infty$ otherwise.

**Example.** Let $X_1 = [[0.2, 0.7]]$ and $X_2 = [[0.5, 0.6]]$ as the DecRects for the $Q_T$ functions for actions $a_1$ and $a_2$ in Figure 1b. Then, $X_{max} = [[0.2, 0.7], [0.5, 0.6]]$. The resulting function is equal to 0.7 for $\delta \leq 0.2$, 0.6 for $0.2 \leq \delta \leq 0.5$ and $-\infty$ elsewhere. Thus, it is the pointwise maximum of the two functions.

## 4.3 COMPUTING NON-DOMINATED ACTIONS (PARETO FRONT)

As can be seen in the DecRect of $V_T$ of the example in Figure 1b shown in Section 4.1, there might be rows that are redundant and can be removed without affecting the function. We call such rows *dominated*, formally defined as follows:

**Definition 2.** *Given a DecRect X with m rows, if $\exists j_1, j_2 \in [m]$ such that $X[j_1] \leq X[j_2]$, $j_1$ is said to be dominated by $j_2$. Hence, the maximal set of the m rows is the set of non-dominated rows and are called Pareto front.*

Remember that $V_t(s_t, \cdot) = \max_{a \in A} Q_t(s_t, a, \cdot)$. That means that the DecRect of $V_t$ is a concatenation of the DecRects of the $Q_t$s. Before concatenation, we annotate the rows of each DecRect with the corresponding action. Then, once we merge them all and remove the dominated rows, if an action has no corresponding rows left in the DecRect of $V_t$, we call that action dominated. Identifying such actions which are not optimal for any $\delta$ is essential for a lot of applications.

Fortunately, Kung et al. (1975) showed that we can compute the maximal set of $m$ $d$-dimensional vectors (here the rows of the matrix), with the natural partial ordering, in $O(m \log_2 m)$ time for $d = 2$ and $d = 3$ and in $O(m(\log_2 m)^{d-2})$ for $d \geq 4$. Thus, computing the non-dominated rows can be done efficiently. This can be done after each computation of a $V_t(s_t, \cdot)$ to remove unnecessary rows and identify dominated actions.

**Example.** Computing the pareto front would remove the row $[[0.3, 0.4]]$ from the DecRect of $V_T$ of Figure 1b.

## 4.4 WEIGHTED SUM OF NIPWC FUNCTIONS

Computing the weighted sum of NIPWC functions with non-negative weights is the essential operation of computing $\mathbb{E}[V_t(s_t, \cdot)]$. Given two piecewise constant non-increasing functions $f_1$ and $f_2 : \Delta \to \mathbb{R}^+$ represented by the DecRects $X_1$ and $X_2$ and two non-negative constants $\alpha_1$ and $\alpha_2$, we describe how to compute the function $f_{sum} = \alpha_1 f_1 + \alpha_2 f_2$ in Algorithm 1.

First, we create an empty DecRect $X_{sum}$. Then, for every pair of rows $x_1 \in X_1$ and $x_2 \in X_2$, we add a new row $x_{sum}$ to $X_{sum}$ where for all $i \leq d$, $x_{sum}[i] = \min\{x_1[i], x_2[i]\}$ (line 5) and $x_{sum}[d+1] = \alpha_1 x_1[d+1] + \alpha_2 x_2[d+1]$ (line 6).

Informally, we compute the pairwise intersections of the hyperrectangles from the two functions and assign them a value of the weighted sum of the two corresponding values.

**Example.** To illustrate the method, we provide two examples: Example 1 is described by Figure 2a and Example 2 by Figure 2b. Figure 2a plots the two single-row DecRects $x_1$ and $x_2$ with $d = 1$ corresponding to two functions as points in the plane and their rectangles (here intervals) in thin lines. $\Delta$ in this example is $\mathbb{R}$ and the rectangles of the two rows are the intervals $[-\infty, 0.2]$ and

---

**Algorithm 1** Weighted Sum of NIPWC Functions Algorithm

---

1: **input:** $X_1 \in \mathbb{R}^{m_1 \times (d+1)}, X_2 \in \mathbb{R}^{m_2 \times (d+1)}, \alpha_1, \alpha_2 \in \mathbb{R}^+$
2: $j_3 \leftarrow 0$
3: **for** $j_1 \in [m_1]$ **do**
4:      **for** $j_2 \in [m_2]$ **do**
5:          $X_{sum}[j_3][i] \leftarrow \min\{X_1[j_1][i], X_2[j_2][i]\}, \forall i \in [d]$
6:          $X_{sum}[j_3][d+1] \leftarrow \alpha_1 X_1[j_1][d+1] + \alpha_2 X_2[j_2][d+1]$
7:          $j_3 \leftarrow j_3 + 1$
8: **return:** $X_{sum}$

---

$[-\infty, 0.8]$ with values $0.7$ and $0.2$, respectively. It also shows the single-row of the resulting DecRect from the weighted sum of the two functions with arbitrary weights $\alpha_1$ and $\alpha_2 \geq 0$. Its interval $[-\infty, 0.2]$ is shown in bold line. Similarly, Figure 2b plots the projected two-single row DecRects $x_1$ and $x_2$ with $d = 2$ of two functions to the plane $\Delta = \mathbb{R}^2$. It shows their rectangles in thin lines. It also shows the projected single row $x_{sum}$ of the weighted summation to $\Delta$ as a point in the plane along with its rectangle which is the intersection of the rectangles and is shown in bold blue lines.

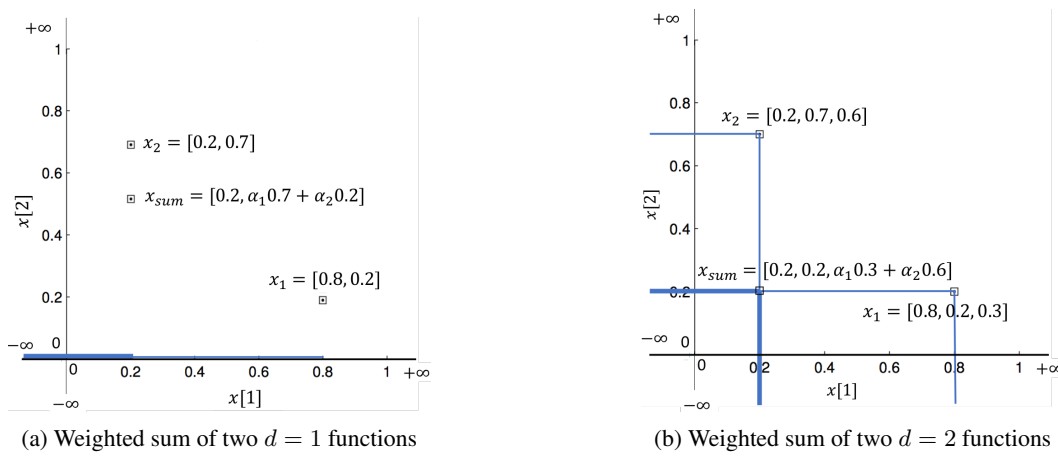

(a) Weighted sum of two $d = 1$ functions        (b) Weighted sum of two $d = 2$ functions

Figure 2: Weighted summation of two NIPWC functions

### 4.5 COMPLEXITY ANALYSIS FOR THE DISCRETE CASE

In this section, we analyze the space and time complexity of the computations done in Section 4. In this section, we assume that the reward vector $r_t$, and consequently the scalarized reward $R_t$, are independent of time and thus we drop their $t$ subscript.

We basically count the number of rows each of the DecRects of the $Q_t$ and $V_t$ has. The $Q_t$s and $V_t$s are combinations of weighted summations and maximizations of the set of size $|S||A|$ of functions $R(s, a, \cdot)$ for different $s \in S$ and $a \in A$. Note that the number of all possible different intersections of combinations of hyperrectangles from a set of $m$ axis-parallel hyperrectangles in $\mathbb{R}^d$ with bottom-left corners at negative infinity is upper bounded by $\left(\frac{m \cdot e}{d}\right)^d$. We prove that in Section B of the Appendix. Recall from Sections 4.2 and 4.4 that the summations and maximizations are done by adding the intersections of the set of hyperrectangles of the $R(s, a, \cdot)$ to the DecRect under construction. Moreover, recall that the Pareto front computation in Section 4.3 removes any row representing a hyperrectangle that is equal or included in another hyperrectangle of a row with higher value. Hence, there are no rows with the same hyperrectangle in any given DecRect. Therefore, the maximum possible number of rows of a DecRect of a $V_t$ or $Q_t$ is upper bounded by $|S||A|$ plus the number of pairwise intersections which is $(|S||A|e/d)^d$. Hence, the space complexity of computing $Q_t$ or $V_T$ is $O(d(|S||A|e/d)^d)$. To compute time complexity, note first that we compute the intersection of two isothetic hyperrectangles with bottom-left corners at negative infinity by computing the minimum of two values in each dimension which requires $O(d)$ time. $V_t$ is the pointwise

maximum of $|A|$ functions each with a DecRect with at most $(|S||A| + (|S||A|e/d)^d)$ rows. This can be done sequentially in $|A| - 1$ steps, one function at a time. At each step computing the pair-wise intersections would take $O\big(d(|S||A|e/d)^{2d}\big)$ time and computing the Pareto front would take $O\big(d(|S||A|e/d)^{2d}\log_2^d(d(|S||A|e/d)^{2d})\big)$ time. Thus, the time complexity of computing $V_t$ given the $Q_t$s is $O\big(d|A|(|S||A|e/d)^{2d}\log_2^d(d(|S||A|e/d)^{2d})\big)$.

$E_{s'|s,a}[V_t(s',\cdot)]$ is a weighted sum of $|S|$ functions each with a DecRect with at most $(|S||A| + (|S||A|e/d)^d)$ rows. We need to multiply the last entry of each row of each DecRect by a constant. That would take $O(|S|(|S||A|e/d)^d)$ time. We then need to add these functions which we do sequentially, adding one function at a time and then compute the Pareto front to remove the dominated rows. Hence, each step would take $O(d(|S||A|e/d)^{2d}\log_2^{d-2}(d(|S||A|e/d)^{2d}))$ time and all $|S| - 1$ steps would take $O(d|S|(|S||A|e/d)^{2d}\log_2^{d-2}(d(|S||A|e/d)^{2d}))$ time, for $d \geq 4$. For $d = 2$ and $d = 3$, the log term has no exponential term. Thus, computing $Q_t$ which is the addition of a function with a single row and $E_{s'|s,a}[V_t(s',\cdot)]$ takes $O(d|S|(|S||A|e/d)^{2d}\log_2^{d-2}(d(|S||A|e/d)^{2d}))$ time.

Once the DecRect of a $Q_t(s, a, \cdot)$ is computed, we decompose the hyperrectangles represented by its rows to disjoint hyperrectangles. Then, retrieving $Q_T(s, a, \delta)$ for any $\delta \in \Delta$ is a matter of finding the enclosing rectangle which takes $O(\log^{d-1} m)$ time as shown by Edelsbrunner & Maurer (1981), where $m$ is the number of rectangles.

From any two rows, we can decompose their hyperrectangles to $O(d)$ disjoint ones in $O(d)$ time while not changing the function. We present the algorithm in Section A of the Appendix. We do that sequentially over the rows in the DecRect of a $Q_t$ by intersecting a row with all previously generated hyperrectangles while not adding already existing rectangles. Instead, we just update their values. This would take $O(d^2(|S||A|e/d)^{4d})$ time. The total number of rows of the resulting DecRect would be $O(d(|S||A|e/d)^{2d})$ since every pair of rows in the original DecRect with $O((|S||A|e/d)^d)$ rows can generate $O(d)$ rows. Moreover, the data structure of Edelsbrunner & Maurer (1981) takes $O(m\log^d m)$ space and preprocessing time. Hence, the space and preprocessing time needed is $O(d^2(|S||A|e/d)^{2d}\log^d d(|S||A|e/d)^{2d})$ and the query time is $O(\log^{d-1} d(|S||A|e/d)^{2d})$.

Therefore, computing $Q_{t-1}$ from $Q_t$ takes polynomial in $|A|$ and $|S|$ and exponential in $d$ time with no dependence on $T$. This is a significant improvement over the doubly exponential bound presented in Lizotte et al. (2012) for linearly scalarized reward. The key difference that allowed this improvement is that the knots (the points of discontinuity) in their case depended on the values of the added or maximized functions and new different knots may be added at each time step even if the reward does not depend on time.

## 5 Value Functions for All Thresholds: Continuous State Space

In this section, we consider the scenario where the state space $S$ is continuous and for any time step $t < T$, action $a_t \in A$ and state $s_t \in S$, the probability distribution of the next state $s_{t+1}$ is a Dirac delta function on some state in $S$. In other words, there is a function $g_t : S \times A \to S$ where $s_{t+1} = g_t(s_t, a_t)$. In this case, Equation (3) will become:

$$Q_t(s_t, a_t, \delta) = R_t(s_t, a_t, \delta) + V_{t+1}(s_{t+1}, \delta), \tag{7}$$

where $s_{t+1} = g_t(s_t, a_t)$.

We will use linear function approximation in computing the $Q_t$ and $V_t$ functions. Formally, for any state $s \in S$, action $a \in A$ and index $i \in [d+1]$, the reward will be expressed as $r_t(s, a)[i] = \psi_s^\intercal \beta_t^a[i]$, where $\psi : S \to \mathbb{R}^c$ maps a state to an $c$-dimensional feature vector and $\beta_t^a[i] \in \mathbb{R}^c$ is a weight vector. This is the same choice taken by Lizotte et al. (2012). This choice assumes separate linear dependence of the reward on the state for each action. We can rewrite Equation (3) to:

$$R_t(s, a, \delta) = \begin{cases} \psi_s^\intercal \beta_t^a[d+1], & \text{if } \forall\, i \leq d,\ \delta[i] \leq \psi_s^\intercal \beta_t^a[i],\ \text{and} \\ -\infty, & \text{otherwise.} \end{cases} \tag{8}$$

Moreover, we assume that the transition functions $g_t$ satisfy $\psi_{s_{t+1}} = F_t^{a_t}\psi_{s_t}$, where $s_{t+1} = g_t(s_t, a_t)$, for some $F_t^a \in \mathbb{R}^{c \times c}$.

As in Section 4, we first discuss in Section 5.1 the data structure in which we store the $Q_t$ and $V_t$ functions. After that, we describe how to find the pointwise scaling, addition and maximization of functions in the new representation in Section 5.2. Finally, in Section 5.3, we discuss how to use the methods of Section 5.2 to compute the $Q_t$ and $V_t$ functions.

## 5.1 ContDecRect Representation of NIPWC functions

DecRect is not suitable for continuous state spaces: we cannot have explicit representations of $Q_t$ and $V_t$ for each $s \in S$. Hence, we introduce a new data structure *ContDecRect* to store the $Q_t$ and $V_t$ functions that handles this issue.

ContDecRect is a data structure that can be used to store functions of the form $f : \mathbb{R}^c \times \Delta \to \mathbb{R}$ that would result from evaluating the Bellman equation, such as $V_t(\cdot, \cdot)$ and $Q_t(\cdot, a, \cdot)$ for some $a \in A$, while having reward as in Equation (8) and fixing $\psi : S \to \mathbb{R}^c$. It is a directed acyclic graph (DAG) with a special structure. For any fixed state $s \in S$ and hence fixed vector $\psi_s$ in $\mathbb{R}^c$, each node of the graph represents an axis parallel hyperrectangle in $\Delta$, with a bottom-left corner at $[-\infty]^d$, and a value in $\mathbb{R}$. The graph consists of several levels. The nodes at level zero have no incoming edges, i.e. no parent nodes, and are called *root* nodes. Any non-*root* node has exactly two parent nodes and an operator of *max* or *sum*. The level of such a node is one plus the maximum of the levels of its parents. Its hyperrectangle would be the intersection of the hyperrectangles of its parents and its value would be either the maximum of the values of its parents or their sum, depending on its operator. Any node can be *ON* or *OFF*. The value of the ContDecRect at certain $\alpha \in \mathbb{R}^c$ and $\delta \in \Delta$ would be the value of the ON node with the maximum level with a hyperrectangle that contains $\delta$. Equivalently, it is the value of the ON node with the maximum value that contains $\delta$. Such a node will be unique for each $\delta$.

Formally, a ContDecRect $\mathcal{H}$ is a tuple: $\langle \mathcal{N}, \mathcal{E}, X, Y, W, O, L \rangle$, where $\mathcal{N}$ is the set of nodes of a DAG with a set of directed edges $\mathcal{E} := \{(n_1, n_2)\}$. $\mathcal{N}$ consists of two disjoint sets of nodes: $\mathcal{N}_0$ of nodes with no incoming edges and $\mathcal{N}_{\bar{0}} = \mathcal{N} \backslash \mathcal{N}_0$ of nodes with exactly two incoming edges. Moreover, $X \in \mathbb{R}^{|\mathcal{N}_0| \times (d+1) \times c}$, $Y \in \mathbb{R}^{|\mathcal{N}_0| \times c}$, $W \in \{\text{ON, OFF}\}^{|\mathcal{N}|}$, $O \in \{max, sum\}^{|\mathcal{N}_{\bar{0}}|}$ and $L \in \mathbb{N}^{|\mathcal{N}|}$. If $(n_1, n_2) \in \mathcal{E}$, where $n_1$ and $n_2 \in \mathcal{N}$, we call $n_1$ a *parent* of $n_2$ and $n_2$ a *child* of $n_1$. The level of a node $n \in \mathcal{N}_0$ is $L[n] = 0$. The level of an $n \in \mathcal{N}_{\bar{0}}$ with parents $n_1$ and $n_2$ is $L[n] = 1 + \max\{L[n_1], L[n_2]\}$. Nodes in $\mathcal{N}_0$ are called *root* nodes. For any node $n \in \mathcal{N}_{\bar{0}}$, its operator $O[n]$ can be *max* or *sum*. $W$ stores the state of each node if it is ON or OFF. The hyperrectangle of a node $n \in \mathcal{N}_0$ is characterized by $X[n]$ which is a matrix in $\mathbb{R}^{(d+1) \times c}$ and its value by the vector $Y[n] \in \mathbb{R}^c$. The semantics of this representation is as follows: for any state $s \in S$ and threshold vector $\delta \in \Delta$, the value of the node is:

$$\begin{cases} \psi_s^\intercal Y[n], & \text{if } \forall\, i \leq d,\ \delta[i] \leq \psi_s^\intercal X[n][i], \text{ and} \\ -\infty, & \text{otherwise.} \end{cases} \tag{9}$$

The value of a non-*root* node $n_3 \in \mathcal{N}_{\bar{0}}$, with a $O[n] = max$ and parents $n_1$ and $n_2$, would be:

$$\begin{cases} \max\{\psi_s^\intercal Y[n_1], \psi_s^\intercal Y[n_2]\}, & \text{if } \forall i \leq d, \delta[i] \leq \min\{\psi_s^\intercal X[n_1][i], \psi_s^\intercal X[n_2][i]\}, \text{ and} \\ -\infty, & \text{otherwise.} \end{cases} \tag{10}$$

If $O[n] = sum$, its value would be $\psi_s^\intercal Y[n_1] + \psi_s^\intercal Y[n_2]$ under the same conditions of Equation (10) and $-\infty$ otherwise. The value of the function would be the value of the unique ON node $n$ with the maximum level with a hyperrectangle that contains $\delta$, i.e. for all $i \in [d + 1]$, $\psi_s^\intercal X[n][i] \leq \delta[i]$.

For each $t \leq T$ and $a \in A$, we represent $R_t(\cdot, a, \cdot)$ as a ContDecRect with a single ON node $n$ with $X[n][i] = \beta_t^a[i]$ for $i \leq d$ and $Y[n][d + 1] = \beta_t^a[d + 1]$. It follows that for all $a \in A$, $Q_T(\cdot, a, \cdot)$ have the same representation.

## 5.2 Scaling, Maximization and Addition of NIPWC functions

Consider two functions $f_1$ and $f_2 : S \times \Delta \to \mathbb{R}$ represented by the ContDecRects $\mathcal{H}_1$ and $\mathcal{H}_2$. We assume that their components are indexed by the same subscripts.

Fix an $\alpha \geq 0$, observe that the function $\alpha f_1$ can be represented by the following ContDecRect:

$$\langle \mathcal{N}_1, \mathcal{E}_1, X_1, \alpha Y_1, W_1, O_1, L_1 \rangle.$$

To find the find the pointwise maximum (sum) of $f_1$ and $f_2$, we create a new ContDecRect that combines both ContDecRects:

$$\mathcal{H}_3 = \langle \mathcal{N}_1 \cup \mathcal{N}_2, \mathcal{E}_1 \cup \mathcal{E}_2, [X_1, X_2], [Y_1, Y_2], [W_1, W_2], [O_1, O_2], [L_1, L_2] \rangle.$$

Then, for every pair of ON nodes $n_1 \in \mathcal{N}_1$ and $n_2 \in \mathcal{N}_2$, we create a child node $n_3$ with a *max* (*sum*) operator and add it to $\mathcal{H}_3$. Formally, $n_3$ gets added to $\mathcal{N}_3$, $(n_1, n_3)$ and $(n_2, n_3)$ gets added to $\mathcal{E}_3$, and $O_3[n_3]$ is set to *max* (*sum*) and $L_3[n_3]$ to $1 + \max\{L_3[n_1], L_3[n_2]\}$. $X_3$ and $Y_3$ would not get updated since the added node is a non-*root* one. $W_3[n_3]$ would be set to ON.

Added nodes represent the intersections of pairs of hyperrectangles from the two functions being maximized (added) and their values are the maximum (sum) of the values of the intersected hyperrectangles. Moreover, their level will be higher than the levels of their parents.

If we are finding the sum of the functions, $W[n]$ would be set to OFF for all $n \in \mathcal{N}_1 \cup \mathcal{N}_2 \subset \mathcal{N}_3$. That is because only the intersections of the hyperrectangles should have the value of the sum, $\delta$s that do not belong to an intersection, at least one of the functions being added is equal to $-\infty$ which means that the sum is also $-\infty$.

The following lemma shows the correctness of this method with a proof in Section C of the Appendix.

**Lemma 1.** *If for any $\delta \in \Delta$, each of $\mathcal{H}_1$ and $\mathcal{H}_2$ has either an ON node that contains $\delta$ with a level that is strictly higher than all other ON nodes that contain $\delta$ or does not have an ON nodes that contain $\delta$, then the constructed $\mathcal{H}_3$ has that property too. Moreover, $\mathcal{H}_3$ would represent the pointwise maximum (sum) of $f_1$ and $f_2$.*

### 5.3 $Q_t$ AND $V_t$ COMPUTATION FOR $t \leq T$

Recall that we represent the $R_t(\cdot, a, \cdot)$s, and consequently the $Q_T(\cdot, a, \cdot)$s, with single-node ContDecRects and $V_T(\cdot, \cdot)$ with the ContDecRect that results from iterative application of the maximization procedure described in Section 5.2 over the $Q_T(\cdot, a, \cdot)$s. Our aim is to construct a ContDecRects that represent $Q_t(\cdot, a, \cdot)$s and $V_t(\cdot, \cdot)$s, i.e. given $\psi_{s_t}$, the hyperrectangles and values of nodes would be specified and given $\delta \in \Delta$ the value of $Q_t$ or $V_t$ would be determined.

Recall from Equation (7) that $Q_t$ is the sum of $R_t$ and $V_{t+1}$. Observe that if we directly apply the addition procedure described in Section 5.2 on the ContDecRects of $R_t$ and $V_{t+1}$, we will need to multiply $\psi_{s_t}$ by $F_t^a$ before applying it to the nodes corresponding to $V_{t+1}$ in the new ContDecRect since $V_{t+1}$ takes $\psi_{s_{t+1}}$ not $\psi_{s_t}$ as input. Instead, we adjust the parameters, specifically $X$ and $Y$, of the ContDecRect of $V_{t+1}$ to account for the multiplication by $F_t^a$.

The upper-right corner of the hyperrectangle and value of a node $n$ of the ContDecRect of $V_{t+1}$ at a state $s_t$ would be $(F_t^a \psi_{s_t})^\intercal X[n][i]$ for $i \leq d$ and $(F_t^a \psi_{s_t})^\intercal Y[n]$, respectively. Note that for any vector $\gamma \in \mathbb{R}^c$, $(F_t^a \psi_{s_t})^\intercal \gamma = (\gamma^\intercal F_t^a) \psi_{s_t}$, since the left hand side is a scalar and thus equal to its transpose. Hence, we update the parameters of the ContDecRect $\mathcal{H}$ of $V_{t+1}$ by updating the $X$ and $Y$ matrices so that $X_{new}[n][i] = X[n][i]^\intercal F_t^a$ and $Y_{new}[n] = Y[n]^\intercal F_t^a$, for each $i \leq d$ and root node $n$ corresponding to action $a$. Using this method, we can get a ContDecRect representation of any $Q_t$.

On the other hand, fortunately, for any $t \leq T$, all of the $|A|$ $Q_t$ functions take $\psi_{s_t}$ as input, as would $V_t$. Hence, to get a ContDecRect representation of $V_t$, one can simply iteratively apply the maximization method of Section 5.2 over the ContDecRects of the $|A|$ $Q_t$ functions.

### 5.4 COMPLEXITY ANALYSIS FOR THE CONTINUOUS CASE

In this section, we analyze the space and time complexity of the computations done in Sections 5.2 and 5.3. As in Section 4.5, we assume that the reward vector $r_t$ and $R_t$ time independent.

We will start by bounding the number of nodes each of the ContDecRects of the $Q_t(\cdot, a, \cdot)$ and $V_t(\cdot, \cdot)$ functions has for each $t \leq T$ and $a \in A$. It is similar to bounding the number of rows of DecRects in Section 5.2.

There are two differences from the complexity analysis in Section 4.5: First, the ContDecRects corresponding to a $Q_t(\cdot, a, \cdot)$ and $V_t(\cdot, \cdot)$ cover all states instead of having different data structure

for each state. Second, because we need to update the parameters of the ContDecRect of $V_t(\cdot, \cdot)$ as we go backward in time.

It follows from the first difference that we have $|A|$ ContDecRects that represent $Q_T$ for all possible inputs and a single ContDecRect to represent in $V_T$ instead of $|A||S|$ DecRects to represent $Q_T$ and $|S|$ DecRects to represent $V_T$, in the discrete case. However, it follows from the second difference that even though the rewards are time-independent, the addition of the reward in the computation of $Q_t$ for $t < T$ introduces a new root node representing a different hyperrectangle than the one that would be added at a different time step.

Still, for any $t \leq T$, we can bound the number of root nodes $\mathcal{N}_0$ and the total number of nodes $\mathcal{N}$ of the ContDecRect of a $Q_t(\cdot, a, \cdot)$ by $O(|A|^{|T|-t})$ and $O(|A|^{2(|T|-t)})$ and those of $V_t(\cdot, \cdot)$ by $O(|A|^{|T|-t+1})$ and $O(|A|^{2(|T|-t+1)})$, respectively. Recall that each $Q_T(\cdot, a, \cdot)$ is represented by a single-node ContDecRect and $V_T(\cdot, \cdot)$ by the ContDecRect resulting from applying the maximization procedure in Section 5.2 over these nodes which would lead to $|\mathcal{N}_0| = |A|$ and $|\mathcal{N}| = 1 + 2 + \cdots + |A| = |A|(|A| + 1)/2 = O(|A|^2)$ nodes. Assume that at $t > 1$, the ContDecRect of each $Q_t(\cdot, a, \cdot)$ has $|\mathcal{N}_0| = O(|A|^{T-t})$ and $|\mathcal{N}| = O(|A|^{2(T-t)})$ and $V_t(\cdot, \cdot)$ has $|\mathcal{N}_0| = O(|A|^{T-t+1})$ and $|\mathcal{N}| = O(|A|^{2(T-t+1)})$. Then, in the construction of the ContDecRect of a $Q_{t-1}(\cdot, a, \cdot)$, one would multiply all the parameters of the root nodes of $V_t$ by $F_{t-1}^a$ and consider them as the root nodes in addition to the one node of $R(\cdot, a, \cdot)$. Thus, it will have $O(|A|^{T-(t-1)})$ root nodes. Moreover, the addition procedure of $R_{t-1}$ and $V_t$ will create a new node for each of the $O(|A|^{2(T-t)})$ nodes of $V_t$ and thus the total number of nodes will be $O(|A|^{2(T-t)})$. Moreover, the set of root nodes of the ContDecRect of $V_{t-1}$ will consists of all the root nodes of the $|A|$ ContDecRects of the $Q_{t-1}$s and thus its size is $O(|A|^{T-(t-1)+1})$. Finally, the total number of nodes of $V_{t-1}$ will be $O(|A|^{2(T-t)})(1 + 2 + \cdots + |A|) = O(|A|^{2(T-(t-1))})$.

Since each of the root nodes store $d + 1$ vectors in $\mathbb{R}^c$ and each non-root node stores the identity of its parents, the space complexity of storing a $Q_t(\cdot, a, \cdot)$ is $O(cd|A|^{T-t} + |A|^{2(T-t)})$ and $V_t(\cdot, \cdot)$ is $O(cd|A|^{T-t+1} + |A|^{2(T-t+1)})$. Moreover time complexity of constructing them is linear in their size, so we have the time complexity the same as the space one. The time complexity of retrieving the value of such functions for a given state $s \in S$ and $\delta \in \Delta$ is also linear in the size since in the worst case every node should be checked if it contains the $\delta$ before reaching the one with the maximum level. However, for a fixed state, all hyperrectangles and values of the nodes would be fixed, which means as in Section 4.5, one can decompose the $O(|A|^{2(T-t)})$ rectangles to $O(d|A|^{4(T-t)})$ disjoint regions. Here, the structure of intersections is explicit: a node hyperrectangle is the intersection of its parents hyperrectangles. One can start from the maximum level assigning the hyperrectangles covered by the ON nodes by the values and removing them from $\Delta$, then decomposing the rest of their parents' hyperrectangles to disjoint hyperrectangles. Then, repeat the process iteratively till the roots. This would take $O(d^2|A|^{8(T-t)})$ time. Once that is done, one can use Edelsbrunner & Maurer (1981) algorithm which would take preprocessing time of $O(d^2|A|^{4(T-t)} \log_2^d d|A|^{4(T-t)})$ and query time of $O(\log_2^d d|A|^{4(T-t)})$.

## 6 CONCLUSION AND FUTURE WORK

We presented a nonlinear reward scalarization function that encodes constraint and goal based specifications. Moreover, we presented data structures that store the $Q$ and value functions that allowed efficient computations of the iterations of the Bellman equation. We presented efficient algorithms to compute a family of policies that cover all preferences. We plan to design an algorithm for learning policies using linear regression over the parameters of the ContDecRects in cases where the transition and reward functions are unknown. Moreover, we plan to design an algorithm to determine dominated actions in the case of continuous state spaces. Finally, we would apply the algorithms to a real life case study.

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

# A  DECOMPOSITION OF THE DIFFERENCE OF TWO HYPERRECTANGLES TO HYPERRECTANGLES

In this section, we provide an algorithm to decompose the complement of a hyperrectangle $r_1 \in \Delta$ with respect to an intersecting rectangle $r_2 \in \Delta$ to $O(d)$ hyperrectangles. The method iterates over the dimensions adding a maximum of two hyperrectangles at each one. It is described in the Algorithm 2.

# B  BASIC FACTS ABOUT INTERSECTIONS OF AXIS-PARALLEL HYPERRECTANGLES

The following lemma bounds the number of different hyperrectangles that may result from the intersection of any combination of $m$ hyperrectangles with bottom left corners at negative infinity in $\mathbb{R}^d$.

**Lemma 2.** *Given $m$ axis-parallel hyperrectangles in $\mathbb{R}^d$, where $d \leq m$, with bottom-left corners at $[-\infty]^d$. The number of different intersections of any combination of hyperrectangles from the $m$ hyperrectangles is upper bounded by $\left(\frac{m \cdot e}{d}\right)^d$.*

---

**Algorithm 2** Decomposition to Hyperrectangles Algorithm

---

1: **input:** $c_{b,1}, c_{u,1}, c_{b,2}, c_{u,2} \in \mathbb{R}^d$
2: $C_{b,new}, C_{u,new} \leftarrow []$
3: **for** $i \in [d]$ **do**
4:     **if** $c_{b,2}[i] > c_{b,1}[i]$ and $c_{b,2}[i] \leq c_{u,1}[i]$ **then**
5:         $c_{b,new}[j] \leftarrow c_{b,1}[j], \forall j$
6:         $c_{u,new}[i] \leftarrow c_{b,2}[i],$
7:         $c_{u,new}[j] \leftarrow c_{u,1}[j], \forall j \neq i$
8:         append $c_{b,new}$ to $C_{b,new}$ and $c_{u,new}$ to $C_{u,new}$
9:         $c_{b,1}[i] \leftarrow c_{b,2}[i]$
10:     **if** $c_{u,2}[i] > c_{b,1}[i]$ and $c_{u,2}[i] < c_{u,1}[i]$ **then**
11:         $c_{b,new}[j] \leftarrow c_{b,1}[j], \forall j \neq i$
12:         $c_{b,new}[i] \leftarrow c_{u,1}[i],$
13:         $c_{u,new}[j] \leftarrow c_{u,1}[j], \forall j$
14:         append $c_{b,new}$ to $C_{b,new}$ and $c_{u,new}$ to $C_{u,new}$
15:         $c_{u,1}[i] \leftarrow c_{u,2}[i]$
16: **return:** $C_{b,new}, C_{u,new}$

---

*Proof.* Since the hyperrectangles are $d$ dimensional with bottom left corner at $-\infty$, the intersection of any number of them is equal to the intersection of at most $d$ of them. A hyperrectangle part of the combination being intersected has to be the minimum in one of the $d$ dimensions to be effecting the intersection. Hence, to generate all possible intersections, one can start by considering one of the $m$ hyperrectangles, and consider the intersection with all combinations of the $m$ hyperrectangles in which it has the minimal upper right coordinate in each dimension. All would have it as the intersection. Then, repeat that for all of the $m$ rectangles. After that, consider any pair of the $m$ hyperrectangles and take their intersection, and then consider all combinations of the $m$ hyperrectangles that would not effect the intersection. Repeat that for all pairs of hyperrectangles. One can continue this process up to considering the intersections of all possible tuples of $d$ hyperrectangles. Hence, the number of possible different intersections of any combination of these $m$ hyperrectangles is upper bounded by $\sum_{i=0}^{d} \binom{m}{i}$. To compute a simple upper bound on this sum, we multiply it first by $\left(\frac{d}{m}\right)^d$ to get:

$$(\frac{d}{m})^d \sum_{i=0}^{d} \binom{m}{i} = \sum_{i=0}^{d} \binom{m}{i}(\frac{d}{m})^d \leq \sum_{i=0}^{d} \binom{m}{i}(\frac{d}{m})^i$$

$$[\text{since } \tfrac{d}{m} \leq 1]$$

$$\leq \sum_{i=0}^{m} \binom{m}{i}(\frac{d}{m})^i < \sum_{i=0}^{\infty} \binom{m}{i}(\frac{d}{m})^i$$

$$[\text{again since } \tfrac{d}{m} \leq 1 \text{ and } m < \infty]$$

$$= \left(1 + \frac{d}{m}\right)^m \leq e^d.$$

Hence, $\sum_{i=0}^{d} \binom{m}{i} \leq \left(\frac{m}{d}\right)^d e^d$. $\qquad\qquad\square$

## C  PROOF OF LEMMA 1

In this section, we provide a proof for Lemma 1 which shows the correctness of the methods used in Section 4.2 to find the pointwise maximum and sum of two ContDecRects. We restate the lemma here for completeness.

**Lemma 1.** *If for any $\delta \in \Delta$, each of $\mathcal{H}_1$ and $\mathcal{H}_2$ has either an ON node that contains $\delta$ with a level that is strictly higher than all other ON nodes that contain $\delta$ or does not have an ON nodes*

*that contain $\delta$, then the constructed $\mathcal{H}_3$ has that property too. Moreover, $\mathcal{H}_3$ would represent the pointwise maximum (sum) of $f_1$ and $f_2$.*

*Proof.* Fix an $s \in S$, then the hyperrectangles and values of the nodes in $\mathcal{H}_1$ and $\mathcal{H}_2$ are fixed. Now, fix $\delta \in \Delta$. If there is no $n_1 \in \mathcal{N}_1$ and $n_2 \in \mathcal{N}_2$ with $W_1[n_1] = W_2[n_2] = $ ON and with hyperrectangles that contain $\delta$, there will be no $n_3 \in \mathcal{N}_3$ that contains $\delta$ and the value of the function will be $-\infty$. If we are finding the pointwise maximum of the functions and there is an ON $n_1 \in \mathcal{N}_1$ with a hyperrectangle that contain $\delta$ but there is no ON $n_2 \in \mathcal{N}_2$ that does, the value of the resulting function would be $f_1(s, \delta)$, and vice versa. However, if we are finding the pointwise sum in this case, the value of the resulting function would be $-\infty$ since $n_1$ (and $n_2$) would be OFF. This is the right value since either $f_1(s, \delta)$ or $f_2(s, \delta)$ would be equal $-\infty$. Finally, if both $\mathcal{H}_1$ and $\mathcal{H}_2$ have ON nodes that contain $\delta$, let $n_1 \in \mathcal{N}_1$ and $n_2 \in \mathcal{N}_2$ be the unique ON nodes with maximum levels in $\mathcal{H}_1$ and $\mathcal{H}_2$ with hyperrectangles that contain $\delta$. Then, the hyperrectangle of the added $max$ ($sum$) node $n_3 \in \mathcal{H}_3$ with parents $n_1$ and $n_2$ will be the unique maximum level ON node that contains $\delta$ in $\mathcal{H}_3$ and its value will be the maximum (sum) of the values of $n_1$ and $n_2$ which are $f_1(s, \delta)$ and $f_2(s, \delta)$. □

