# OpenReview forum: "Multi-Objective Value Iteration with Parameterized Threshold-Based Safety Constraints"
_ICLR.cc/2019/Conference_

### Official Review · AnonReviewer2 · 2018-10-28
**Interesting first step, but hard to follow and no practical demonstrations.**

**Rating:** 3
**Confidence:** 4

**Review:**

Summary

The authors consider RL with safety constraints, which is framed as a multi-reward problem. At a high-level, this involves finding the Pareto front, which optimally trades off objectives. The paper primarily introduces and discusses a discretization scheme and methods to model the Q-value function as a NIPWC (non-increasing piecewise constant function). NIPWC are stored as values over discrete partitions of state-action spaces. To do so, the authors introduce two data structures DecRect and ContDecRect to store Q function values over geometric combinations of subsets of state-action space.

The authors discuss how to execute elementary operations on these data structures, such as computing max(f(x), g(x)), weighted sums, etc. The goal is to use these operations to compute Bellman-type updates to compute optimal value/policy functions for multi-reward problems. The authors also present complexity analysis for these operations.

Pro
- Extensive discussion and analysis of discrete representations of Q-functions as NIPWCs.

Con
- A major issue with this work is that it is very densely written and spends a lot of time on developing the discretization framework and operations on NIPWC. However:
- There is no clear practical algorithm to solve (simple) multi-reward RL problems with the authors' approach.
- No experiments to demonstrate a simple implementation of these techniques.
- Even though multi-reward settings are the stated problem of interest, authors don't discuss Pareto front computations in much detail, e.g., section 4.3 computing non-dominated actions is too short to be useful.
- The discussion around complexity upper bounds is too dense and uninsightful. For instance, the bounds in section 5 all concern bounds on the Q-value as a function of the action, which results in upper bounds as a function of |A|. But in practice, the action is space is often small, but the state space is high-dimensional. Hence, these considerations seem less relevant.

Overall, this work seems to present an interesting computational scheme, but it is hard to see how this is a scalable alternative. Practical demonstrations would benefit this work significantly.

Reproducibility
N/A

---

> ### Author Response · Authors · 2018-11-08
> **Thank you for your time reading the paper and giving feedback**
>
> - This paper makes theoretical contributions with (a) developing value iteration algorithms for known MDPs with discrete and continuous state spaces that generate policies for all parameters for a parameterized reward and (b) in providing complexity bounds (see the discussion above regarding bounds).
>
> - For the discrete part, the Pareto front computations can be done using off-the-shelf Pareto front computation algorithm as mentioned Section 4.2. Is there anything specific about the method that is unclear?
>
> - We will work on simplifying the discussion of the bounds. The state space size only affects the bound by the dimension of the feature space c (first line of the last paragraph of Section 5.4). Moreover, the last paragraph (starting from “However, for a fixed state,…") discusses the complexity for a fixed state. We will try to simplify presentation in general. If you can point to specific parts that are unclear, please let us know, that would help too.
>
> - For the continuous part, we have an efficient method now that would appear in a later work.
>
> - The RL extension of the work is planned future work. This would go roughly like this: start with offline data/trajectories as in (Lizotte et al. 2010 and 2012); learn the reward functions for the reward vector components and the transition probabilities of the MDP from data and then apply our algorithm for this MDP.
>
> - We acknowledge that experimental evaluations are going to be important; they are in the works; however, we believe that they are somewhat orthogonal to the contributions claimed in the current submission.
>
> Please if you have any further comments or questions let us know.

---

### Official Review · AnonReviewer1 · 2018-11-02
**why is this an important problem?**

**Rating:** 5
**Confidence:** 2

**Review:**

The authors provide an algorithm that aims to compute optimal value functions and policies as a function of a set of constraints.  The ideas used for designing the algorithm seem reasonable.  However, I don't fully understand the motivation here.  Given a set of constraints, one can simply carry out value iteration with what the authors call the scalarized reward in order to generate an optimal policy.  Why go through the effort to compute things in a manner parameterized by the constraints?  Perhaps the intention is to use this for sensitivity analysis, though the authors do not discuss that?

---

> ### Author Response · Authors · 2018-11-08
> **Thank you for your time reading the paper and giving feedback**
>
> We replied to the general audience above as this is an important question and we thought it should be a general comment.

---

> > ### Comment · AnonReviewer1 · 2018-11-23
> > **I better understand the motivation now**
> >
> > The author's response was useful to understanding what they view as important about the problem.  I think its an interesting problem, but the authors should revise to make a clear case for the significance and offer examples of contexts where the approach is practical and serves an important need.  Such a context needs to be one where the complexity of the algorithm is not prohibitive while that of solving for each agent individually would render the trivial alternative impractical.

---

### Official Review · AnonReviewer4 · 2018-11-19
**Interesting direction, but clarity should be improved**

**Rating:** 5
**Confidence:** 4

**Review:**

I generally like the paper. The paper discussed a constrained value iteration setting where the safety contraints must be greater some threshold, and thresholds \delta are parameters. The paper attempts to develop an value iteration algorithm to compute a class of optimal polices with such a parameter. The algorithm is mainly based on a special design of representation/data structure of PWC function, which can be used to store value functions and allows to efficiently compute several relevant operations in bellman equation. A graph-based data structure is developed for continuous state domains and hence value iteration can be extended to such cases.

In general, the paper presents an interesting direction which can potentially help solve RL problems with the proposed constraint setting. However, the paper spends lots of effort explaining representations, but only a few sentences explaining about how the proposed representations/data structures can help find a somehow generic value iteration solution, which allows to efficiently compute/retrieve a particular solution once a \delta vector is specified. The paper should show in detail (or at least give some intuitive explanations) that using the proposed method can be more efficient than solving a value iteration for each individual constraint given that the constraints are independent. Specifically, the author uses the patient case to motivate the paper, saying that different patients may have different preferred thresholds and it is good to find class of policies so that any one of those policies can be retrieved once a threshold is specified. However, in this case, when dealing with only one patient, the dimension of reward is reduced to 1 (d = 1), while the computation of the algorithm is exponential in d, plus that the retrieval time is not intuitive to be better, so it is unsure whether computing such a class of policies worth.

In terms of novelty, the scalarization method of the vector-valued reward seems intuitive, since breaking a constraint means a infeasible solution. Furthermore, it is also unclear why the representation of PWC in discrete case is novel. A partial order on a high-dimensional space is naturally to be based on dominance relation, as a result, it seems natural to store value function by using right-top coordinates of a (hyper)rectangle.

As for the clarity, though the author made the effort to explain clearly by using examples after almost every definition/notation, some important explanations are missing. I would think the really interesting things are the operations based on those representations. For example, the part of computing summation of two PWC function representation is not justified. Why the summation can be calculated in that way? Though the maximum operation is intuitive, however, the summation should have some justification. I think a better way to explain those things is to redefine a new bellman operator, which can operate on those defined representations of PWC function.

I think it could be a nice work if the author can improve the motivation and presentation. Experiments on some simple domains can be also helpful.

---

> ### Author Response · Authors · 2018-11-26
> **Thank you for your time reading the paper and giving feedback**
>
> - To avoid confusion, d is the dimension of the parameter space and is not related to the number of users. But yes, for a single patient, the parameters will be fixed and the reward vector will be scalarized. We added a comment below about the motivation of the work after a question from reviewer1. I hope that would show why this work is important.
>
> - Remember that we compute the sum by assigning the intersection of every pair of hyper rectangles, from the two functions being added in the space [0,1]^d, the sum of the values associated with these hyper rectangles. The parts of the domain that do not belong to the intersection will have negative infinity value since at least one of the rectangles will have negative infinity value there. Also, recall that the value of a function at a point in the parameter space is the maximum of the values associated with the enclosing rectangles. At any point in the parameter space, the sum of the max values of the hyper rectangles enclosing the point from both functions will be the max of the sums of any two pairs of values of enclosing hyper rectangles since the values are all non-negative. Hence, every point in the domain/parameter space will get a value equal to the sum of the values of the two functions at that point.
>
> - We acknowledge that experimental evaluations are going to be important; they are in the works; however, we believe that they are somewhat orthogonal to the contributions claimed in the current submission.

---

### Author Response · Authors · 2018-11-08
**Motivation of the work and main contributions**

Recall, the problem addressed in our paper  is to compute the optimal value functions for a family of constraints/thresholds. Recall also for the following discussion S is the state space, A is the action set, d is the dimension of the reward vector, and T is the time horizon.

1. Why this is an important problem? Two reasons:

(a) Computing optimal policy for a family: A family of users with different preferences need to be served according to their respective optimal policies, with a (parameterized) policy that is computed once and for all. This arises when the computation cost of for the optimal policy is greater than the retrieval cost from parameterized policy computed offline.
     (i) Computing a single policy takes O((d + |S|)|S||A| T)  time in the discrete time case (|S| |A| Q-functions to compute at each of the T time steps. Computing each Q-function takes O(d) time to compute the reward and O(|S|) time to compute the expectation). Retrieving a policy takes O(T log^{d-1} (d(|S||A|e/d)^{2d})) time as shown in Section 4.5.
    (ii) Computing a single policy in the continuous case takes O((c d |A|)^T)), where c is the dimension of the state feature space. On the other hand, our retrieval time of a policy is O(T log^d (2 d|A|^{4T})).

(b) Sensitivity analysis: Consider a common situation where the user’s is not absolutely sure about her preferences as represented by a specific threshold. She needs to know how sensitive is the optimal action to her current preference. If the optimal action changes and have significantly better reward with a small change of preference, she might changes her preference.


A concrete instance. Revisiting the example from paper, different patients have different preferences on the thresholds on side effects. Accurately choosing a threshold based for an individual would be hard (this depends on many personal factors). Our algorithm provides the range of thresholds that makes an action (medication) an optimal choice. Our method will give more information to the patient on how sensitive the optimal action is to the particular threshold she choses.

Moreover, it is more computationally efficient (when d  is small relative to T, |S| and |A|) to compute the family of policies for all preferences and only retrieve the optimal one based on an incoming patient preference rather than computing one for every incoming patient.

2. Exponential improvements in scalability.
The complexity bounds for synthesizing the family of policies achieved is exponential in d and polynomial in |S| and |A| and linear in T;  In (Lizotte et al 2012), for the discrete case, only the case analyzed was for d=2, and the bound is T(|S||A|)^T, which is exponential in T. For the continuous case, our bound is exponential in T while that of (Lizotte et al 2012) for learning in the continuous case is doubly exponential in T.

---

### Meta-Review · Area_Chair1 · 2018-12-10
**Clarity and motivation need improvement**

**Confidence:** 4
**Recommendation:** Reject

**Metareview:**

The main issue with the work in its current form is a lack of motivation and some clarity issues. The paper presents some interesting ideas, and will be much stronger when it incorporates a more clear discussion on motivation, both for the problem setting and the proposed solutions. The writing itself could also be significantly improved.